# Deep Neural Network Utilizing Remote Sensing Datasets for Flood Hazard Susceptibility Mapping in Brisbane, Australia

Bahareh Kalantar [1,*], Naonori Ueda [1], Vahideh Saeidi [2], Saeid Janizadeh [3], Fariborz Shabani [4], Kourosh Ahmadi [5] and Farzin Shabani [6,7]

1   RIKEN Center for Advanced Intelligence Project, Goal-Oriented Technology Research Group, Disaster Resilience Science Team, Tokyo 103-0027, Japan; naonori.ueda@riken.jp
2   Department of Mapping and Surveying, Darya Tarsim Consulting Engineers Co. Ltd., Tehran 14578-43993, Iran; saeidi@daryatarsim.com
3   Department of Watershed Management Engineering, College of Natural Resources, Tarbiat Modares University, Tehran 15119-43943, Iran; janizadehsaeid@modares.ac.ir
4   Department of Civil Engineering, Kermanshah Azad University, Kermanshah 67189-97551, Iran; fariborz.shabani1977@gmail.com
5   Department of Forestry, Faculty of Natural Resources and Marine Sciences, Tarbiat Modares University, Tehran 15119-43943, Iran; kourosh.ahmadi@modares.ac.ir
6   Global Ecology and ARC Centre of Excellence for Australian Biodiversity and Heritage, College of Science and Engineering, Flinders University, GPO Box 2100, Adelaide, SA 5001, Australia; farzin.shabani@flinders.edu.au
7   Department of Biological Sciences, Macquarie University, Sydney, NSW 2109, Australia
*   Correspondence: bahareh.kalantar@riken.jp; Tel.: +81-362252482

**Abstract:** Large damages and losses resulting from floods are widely reported across the globe. Thus, the identification of the flood-prone zones on a flood susceptibility map is very essential. To do so, 13 conditioning factors influencing the flood occurrence in Brisbane river catchment in Australia (i.e., topographic, water-related, geological, and land use factors) were acquired for further processing and modeling. In this study, artificial neural networks (ANN), deep learning neural networks (DLNN), and optimized DLNN using particle swarm optimization (PSO) were exploited to predict and estimate the susceptible areas to the future floods. The significance of the conditioning factors analysis for the region highlighted that altitude, distance from river, sediment transport index (STI), and slope played the most important roles, whereas stream power index (SPI) did not contribute to the hazardous situation. The performance of the models was evaluated against the statistical tests such as sensitivity, specificity, the area under curve (AUC), and true skill statistic (TSS). DLNN and PSO-DLNN models obtained the highest values of sensitivity (0.99) for the training stage to compare with ANN. Moreover, the validations of specificity and TSS for PSO-DLNN recorded the highest values of 0.98 and 0.90, respectively, compared with those obtained by ANN and DLNN. The best accuracies by AUC were evaluated in PSO-DLNN (0.99 in training and 0.98 in testing datasets), followed by DLNN and ANN. Therefore, the optimized PSO-DLNN proved its robustness to compare with other methods.

**Keywords:** deep learning neural network; flood susceptibility mapping; particle swarm optimization; Australia

## 1. Introduction

It has become commonplace to say that destructive flood hazards are reported widely and globally. Excessive urbanization and climate change are more often blamed as the main reasons for such hazards [1,2]. The massive human, economic, and infrastructure losses resulting from flood occurrences necessitate flood management, prediction, and early warning systems [3,4]. At a global scale, in the period 1995–2015, it was reported that 109 million people were influenced by flood hazards, with annual direct costs of

75 billion dollars [5]; only between 2011–2012, indirect damages and losses were reported as 95 billion dollars [6]. The annual human life losses due to flooding are estimated to be 20,000 people [5]. This has led to the creation of some national flood mapping agencies and portals around the world, such as the USA (www.fema.gov, accessed on 30 May 2021), Ireland (www.floodinfo.ie, accessed on 30 May 2021), and Australia (www.ga.gov.au, accessed on 30 May 2021), providing a database of flood studies, maps, metadata, weather warnings alert, and flood risk maps. Flood research, especially in the Queensland government and local government areas of Australia, is one of the extensive studies to manage the Brisbane river floodplain. According to historical devastating flood events (1893, 1974, and 2011) in the Brisbane basin, annual exceedance probability of 55.3% was estimated as the chance of flood occurrences with experiencing at least one in an 80-year lifetime (https://cabinet.qld.gov.au/documents/2017/Apr/FloodStudies/Attachments/Overview.pdf, accessed on 30 May 2021), causing the emergency and timely response in the region. However, the mapping of potential hazardous areas remains unsettled. Certainly, prediction of probable zones for flood hazards and identification of the likelihood of flood occurrence as susceptibility mapping might assist the decision makers in timely flood mitigation, early warning, and decreasing the damages [7,8]. Precise monitoring, response, and urban management by the city planners require advanced technologies and large geospatial datasets [9]. Today, remote sensors provide big data from the entire globe in no time. Such data prove their effectiveness in sustainable urban and environment management and the creation of urban informatics for better data representation, visualization, and interpretation of new information [10,11]. Along with big data collection, fast and accurate analysis and visualization are vital [10]. Hence, data mining, modeling, and developing robust and accurate algorithms obtain more attention, especially for natural disaster management and urban planning [12,13]. In this context, the integration of remote sensing (RS) technologies and geographic information system (GIS) tackles the spatial, temporal, and regional challenges of flood processes, and the availability of various earth observation data helps to predict and map the flood events and susceptible areas [1,3,14,15]. Therefore, we tried to explore and exploit artificial and deep learning neural networks and proposed optimization to obtain higher accuracy. The effect of robust big data mining and modeling on the reliability of the flooded zone prediction as well as determining the most important conditioning factors for this hazard in the subtropical area will reveal new guidelines for the authorities to plan for effective flood management.

*Related Studies*

A lot of studies have been carried out on flood modeling and susceptibility mapping [16–20], while the choice of appropriate flood conditioning factors and more accurate and certain algorithms is still under investigation. Chen et al. [15] divided the two common groups of algorithms for flood modeling and mapping into qualitative and quantitative methods. It was stated that exploiting statistical and probabilistic models is the main focus of the quantitative methods such as weights-of-evidence (WOE), frequency ratio (FR), and logistic regression (LR). Some examples of qualitative models are analytic network process (ANP) and analytic hierarchy process (AHP) [16]. However, the dynamic nature and complexity of the flood events in a large-scale region provoke the use of more robust models where linear and simple statistical methods seem unreliable [1,19]. In research performed by Dano et al. [16], the intergradations of GIS and ANP for flood prediction and mapping were investigated. However, the calculation of the relative weights of flood conditioning factors was dependent on expert knowledge and questionnaires due to the ANP mathematical model. Although the proposed model exhibited simple procedures, its dependency on expert opinion makes it incompatible with quantitative methods and inapplicable in the broad area [19].

Recently, machine learning (ML) algorithms (e.g., random forest (RF), support vector machine (SVM), decision tree (DT), and artificial neural networks (ANN)) and optimized models proved their abilities to handle large numbers of variables and large datasets timely

and accurately [1,4,21]. ML algorithms have been successfully applied in many applications such as landslide, flood, and wildfire susceptibility mapping [13,19,22,23]. However, the implementation and reliability of these methods still need further investigation in natural hazard prediction [15,19].

Khosravi et al. [24] applied some multiple-criteria decision-making algorithms (MCDM) and ML algorithms (e.g., naive Bayes (NB) and naive Bayes tree (NBT)) to predict and locate the areas prone to flooding and report the outperformance of the ML algorithms. Accordingly, the altitude was mostly responsible for the flood events, whereas land surface curvature represented no influence. To predict the inundation area, Kia et al. [25] applied ANN using seven conditioning factors and concluded that the most significant and insignificant factors influencing the flood in the area were elevation and geology, respectively. Again, ANN was exploited by Falah et al. [26] to determine the flood susceptible areas using five factors, and it was highlighted that drainage density was the most and elevation was the least important factor in the region. Their results were assessed by the area under curve (AUC), and the values of 94.6% and 92.0% were obtained for training and validation, respectively. In another study [27], the implementation of ANN and soil conservation service runoff (SCS) with seven conditioning factors was investigated. The best root mean square error (RMSE) was acquired by ANN as 0.16 at peak flow, promoting precipitation and normalized difference vegetation index (NDVI) as the most influential factors. The results of these studies also emphasized the case-specific selection of the flood causative factors according to the region and its conditions [26]. However, the last three studies did not comprehensively explore and compare ANN performance with other popular methods. Hence, exploiting ANN might comprehensively examine its robustness to compare with other models in different sites, and it can be a suitable benchmark to evaluate the proposed ensemble algorithm.

Although ANN performs well in flood susceptibility mapping, it is limited to one or two hidden layers for the optimization and complex problems. Therefore, deep learning methods with multilayer architectures and higher performance and accuracy are in demand [28]. The AUC values better than 0.96 were recorded by [29] using a deep learning neural network (DLNN) to predict flash flood susceptibility, and this model outperformed multilayered perceptron neural network (MLP-NN) and SVM and proved to be a superior model for the GIS dataset in the study area. In this regard, the optimization and ensemble models also proved to be practical and effective to obtain more certainty and accuracy during the modeling [1,17,19,30,31]. Bui et al. [22] also predicted flash flood zones in tropical areas using optimized DLNN with four swarm intelligence algorithms (e.g., grasshopper optimization algorithm (GOA), social spider optimization (SSO), grey wolf optimization (GWO), and particle swarm optimization (PSO)). Their proposed methods exhibited higher accuracies than individual benchmarks such as PSO, SVM, and RF. However, the last two study areas suffer from flash floods that might be a consequence of heavy and intensive precipitation. Therefore, careful examination of other conditioning factors for flood susceptibility mapping is desired.

The ensemble of models reportedly improved the performance of the predictions. Tehrany et al. [32] compared FR, SVM, and their ensemble models within a dataset including digital elevation model (DEM), slope, geology, curvature, river, stream power index (SPI), land use/cover, rainfall, topographic wetness index (TWI), and soil type to map the susceptible areas for flood, and they established the better performance of the proposed ensemble model. In another study, Tehrany et al. [1] introduced a GIS-based ensemble method (evidential belief function and SVM with linear, polynomial, sigmoid kernel, and radial basis functions) for susceptibility mapping in the Brisbane catchment, Australia, using the 12 conditioning factors of altitude, slope, aspect, SPI, TWI, curvature, soil type, land use/cover, geology, rainfall, distance from road, and distance from river. The authors reached the maximum accuracies by the ensemble models (e.g., 92.11%) up to 7% higher than the individual algorithms. Another work [15] conducted an investigation into ensemble-based machine learning techniques by deploying reduced-error pruning

trees (REPTree) with bagging (Bag-REPTree) and random subspace (RS-REPTree) using 13 flood-influencing factors to estimate the probability of the flood zones. Their experiment also ranked the ensemble model's performance as superior compared with the individual models. Four models, namely FR, the ensemble of FR and Shannon's entropy index (SE), the ensemble of the FR and LR, and the statistical index (SI) model, were exploited by Liuzzo et al. [18], and 10 factors affecting floods were included in the modeling. The highest performance was reported by the ensemble of the FR and LR, again.

The optimization was widely reported in recent years. Sachdeva et al. [19] proposed an optimized model of PSO and SVM and compared its result with susceptibility maps from RF, neural networks (NN), and LR. They used 11 conditioning factors, namely elevation, slope, aspect, TWI, SPI, plan curvature, soil texture, land cover, rainfall, NDVI, and distance from rivers. Their findings highlighted the lowest (91.86%) and highest (96.55%) accuracies for the LR and optimized model, respectively. Li et al. [31] applied a discrete PSO-based sub-pixel flood inundation mapping (DPSO-SFIM) algorithm to create flood maps and reported the success of optimization compared with the other four models. Adaptive neuro-fuzzy inference systems (ANFIS) with three optimization algorithms (ant colony optimization, genetic algorithm, and PSO) were studied by [30], and the susceptible areas for flood were accurately mapped by ANFIS-PSO. In a study of flooded area mapping [33], the authors used synthetic aperture radar (SAR) data and the interferometric SAR information about a flood hazard and improved the performance of a Fuzzy C-Means model by integration and optimization with PSO. Similarly, a model enhancement was reported by [34] by the integration of PSO with Bayesian regulation back propagation NN using Landsat images of the flood events. Nevertheless, the focus of both works was on flood mapping and not flood susceptibility mapping. Thus, the applicability of optimized PSO-DLNN models in flood susceptibility mapping in different sites still needs more consideration, and testing the capabilities and level of improvement of the optimization method to map the susceptible flood zones was also the motivation to conduct this research.

Flood susceptibility mapping of the Brisbane catchment, Australia, using other algorithms (SVM, evidential belief function, LR, and FR) was practiced by [1,15,16], and at best, higher performance accuracy of 92.11% was obtained. Then, we looked for a way to achieve better performance. To the best of the authors' knowledge, there is no comprehensive study fully exploring the optimized DLNN via PSO to map flood susceptibility in the Brisbane catchment, Australia. Therefore, we aimed (1) to classify the flood susceptible zones in the study area into five probability classes (i.e., very low, low, moderate, high, and very high) using three models, namely ANN, DLNN, and the optimized DLNN using PSO (PSO-DLNN); (2) to assess and compare the accuracy and reliably of the three models based on sensitivity, specificity, the area under curve (AUC), and true skill statistic (TSS) tests; and (3) to determine the most important factors (i.e., altitude, slope, aspect, curvature, distance from river, distance from road, rainfall, land use, lithology, soil, SPI, TWI, and sediment transport index (STI)), influencing the flood occurrence, in the subtropical climate region.

## 2. Study Area and Materials

### 2.1. Study Area

The region of study (Figure 1) is a part of the Brisbane river catchment in Queensland, Australia. As a result of tropical cyclones, two flood events were recorded on February 1893 and January 1974 (the largest flood of 20th century) based on Queensland Government reports. Brisbane experienced devastating flood hazards between 2010 and 2011 [20] and, later on, in January 2013. Therefore, the region is considered as a floodplain area and is under continuous study by the Queensland government in partnership with other sectors. Here, the study area (Figure 1) covers 1,474,617 square meters and lies between latitudes of 27°45′S and 27°25′S and longitudes of 152°50′E–153°05′E. The majority of the area represents the dense urbanization. The altitude in the region varies from 0 to 548 m, with humid subtropical climate characteristics, average temperature of 20.3 °C, and annual

rainfall of 1168 mm [1]. According to the geology, Neranleigh-Fernvale beds, clay, and sandstone are the dominant coverage (Table 1 and Figure 2i).

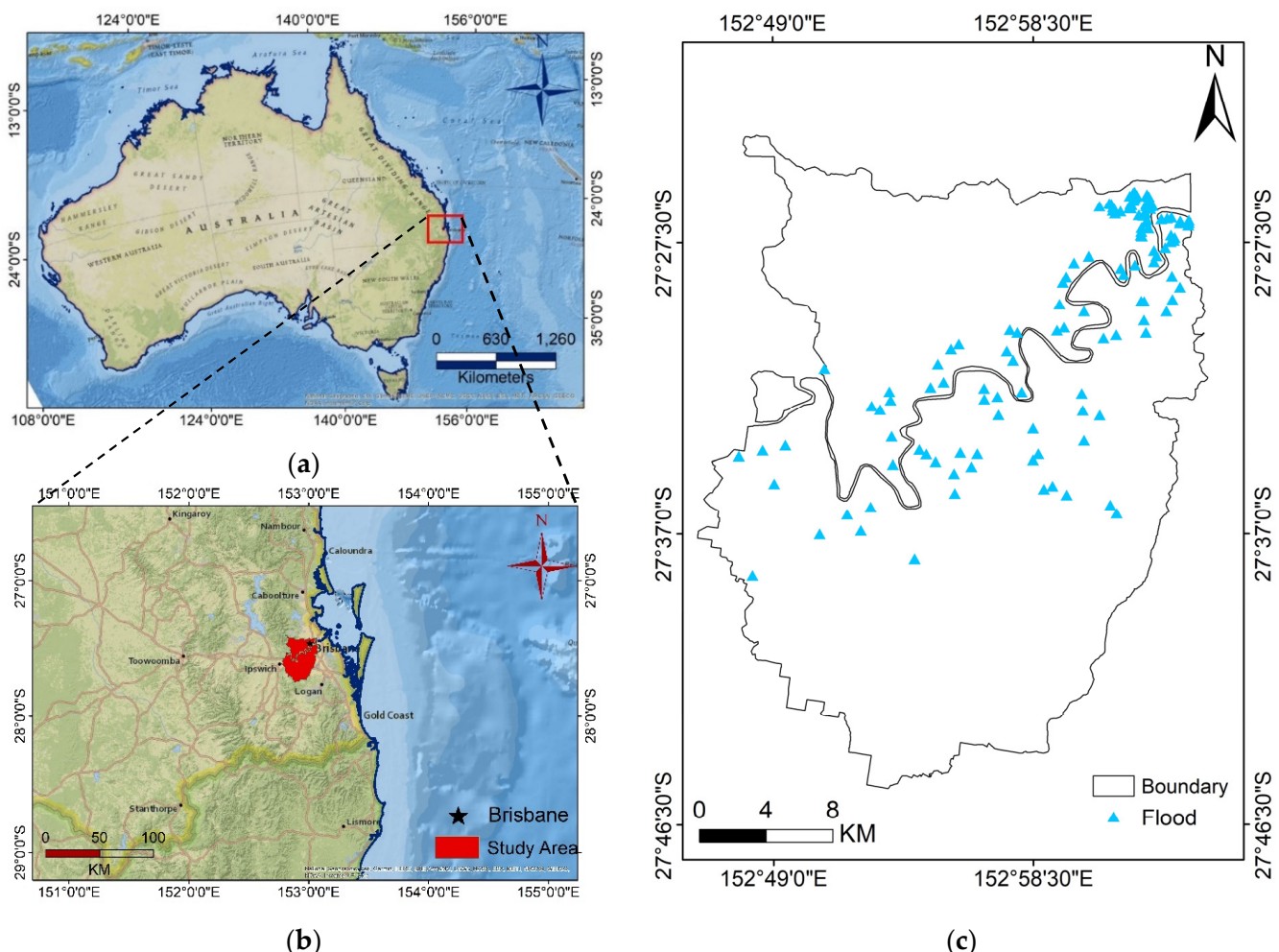

**Figure 1.** General location of the study area: (**a**) Australia map; (**b**) location map of study area; (**c**) the study area including flood and non-flood inventory points.

**Table 1.** Lithology contents and classes in the study area.

| Code | Label | Name |
|:---:|:---:|:---:|
| 1 | | Bellthorpe andesite, Brookfield volcanics, Gilla volcanics, unnamed volcanic units |
| 2 | | Bundamba Group (i.e., Marburg Subgroup and Woogaroo Subgroup) and Landsborough sandstone |
| 3 | | Ipswich coal measures |
| 4 | | Middle to Late Triassic volcanic units, southeast Queensland |
| 5 | | Neranleigh-Fernvale beds, Bunya phyllite |
| 6 | | Paleocene–oligocene sediments |
| 7 | | Quaternary alluvium and lacustrine deposits |
| 8 | | Td-QLD |
| 9 | | Triassic intrusives in south-eastern and central Queensland |

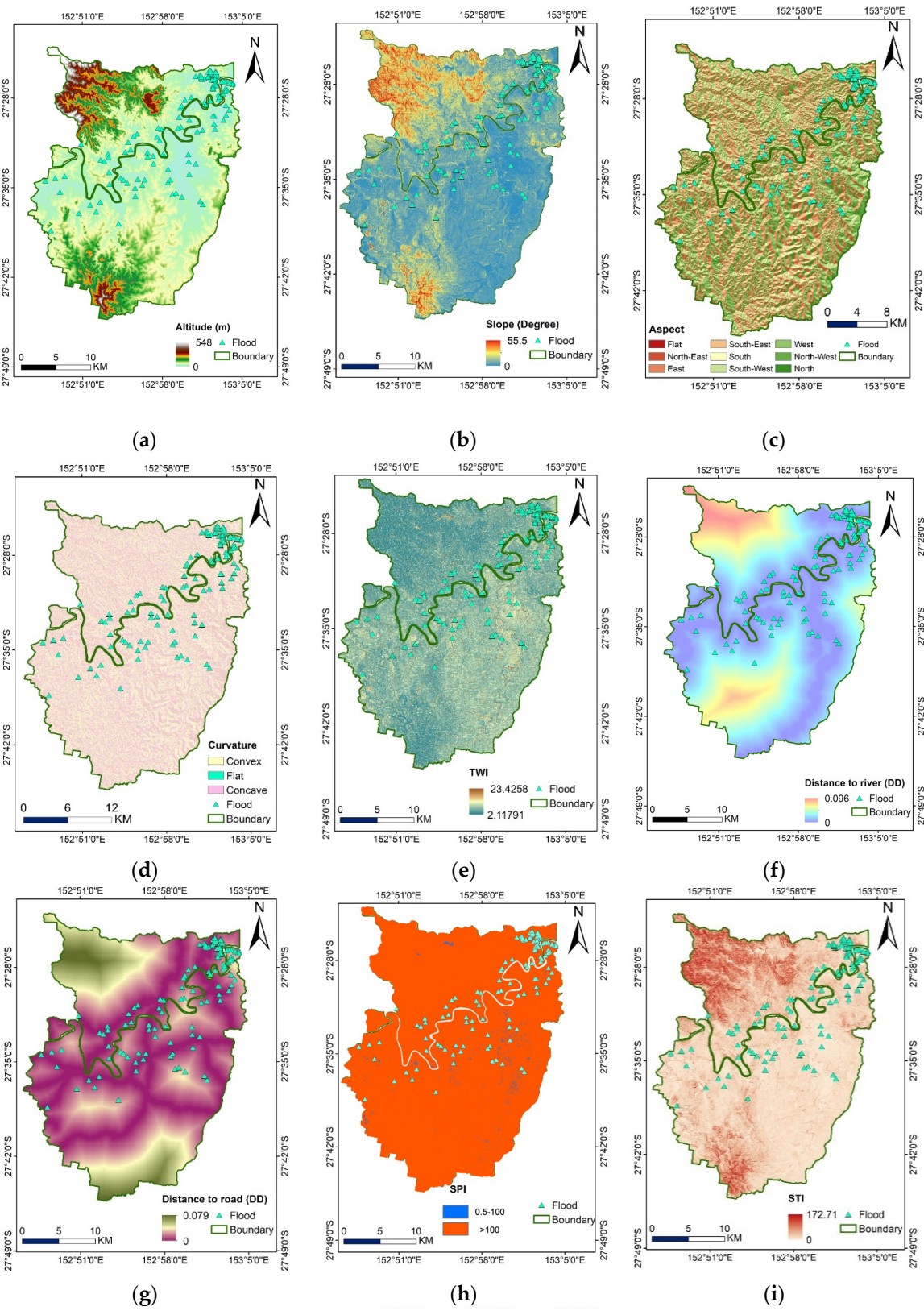

**Figure 2.** *Cont.*

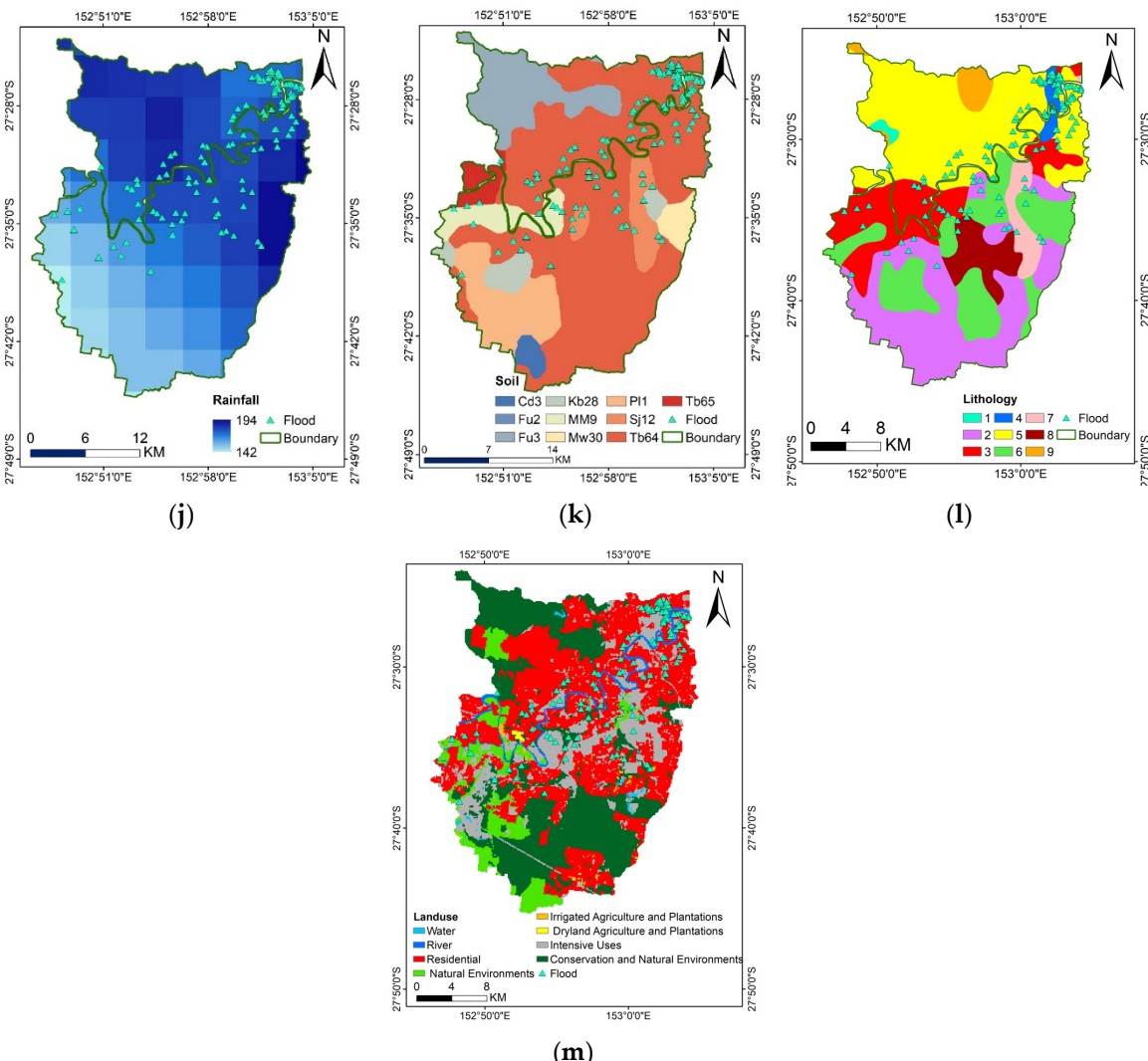

**Figure 2.** Thirteen explanatory factors for model development in this study: (**a**) altitude, (**b**) slope, (**c**) aspect, (**d**) curvature, (**e**) distance to river, (**f**) distance to road, (**g**) SPI, (**h**) TWI, (**i**) STI, (**j**) total annual rainfall, (**k**) soil, (**l**) lithology, and (**m**) land use.

### 2.2. Data Description

For the purpose of flood modeling and susceptibility mapping, two types of datasets were prepared: an inventory map of the past flood events and the factors influencing this hazard [20].

### 2.2.1. Flood Inventories

The flood inventory consisted of 128 historical areas from Brisbane floods dated back to 2001 and extracted from the high-resolution aerial photography after the flood event (Figure 1c). It was provided by Queensland government open data portal (https://www.data.qld.gov.au/dataset/flood-extent-series, accessed on 30 May 2021). At the time of data preparation, we randomly selected non-flood points (128) within the area using a precise buffer around the flood zones by binary classification of flood and non-flood locations using ArcGIS software. Thereafter, the inventory map was randomly divided into 70% and 30% points to extract the corresponding values of 13 conditioning factors for the training and testing of the models, respectively.

### 2.2.2. Explanatory Factors

According to the literature [1,17,24,26,30,32] and availability of the data in the region, 13 causative factors, namely altitude, slope, aspect, curvature, distance from river, distance

from road, rainfall, land use, lithology, soil, SPI, TWI, and STI, were selected. Light detection and ranging (LiDAR) data from the Australian government (http://www.ga.gov.au/elvis/, accessed on 30 May 2021) were the source of a DEM in 10-m resolution to create flood conditioning factors such as altitude, slope, aspect, curvature, SPI, TWI, and STI.

The Topographic Factors

The topographic factors such as altitude, slope, aspect, curvature, and TWI play major roles in predicting the regions of flood occurrence [1]. The altitude of the region was derived from DEM varying from 0 to 548 m (Figure 2a). Accordingly, the slope of the study area (Figure 2b) was calculated between 0 and 55.5 °, and this steep area increases the runoff velocity [1]. The aspect map (Figure 2c) reflects the direction of the terrain and then the direction of floodwater flow [29], and it was classified into nine classes, namely flat, northeast, east, southeast, south, southwest, west, and northwest. The curvature (Figure 2d) indicates the divergence and convergence of runoff, and it was categorized into three classes of convex, flat, and concave [29,32]. TWI (Figure 2e) measures the direction and accumulation of water flow due to the gravity of the place [32]. It is calculated by Equation (1):

$$\text{TWI} = \ln\left(\frac{A_s}{\tan\beta}\right) \tag{1}$$

where $A_s$ is the upslope area per unit of contour length (m$^2$/m) and $\beta$ measures the topographic gradient or local slope gradient in degree [35]. The higher value indicates higher accumulation and runoff flow and thus more sensitivity to the flood occurrence [3,7].

Moreover, distance to river and road (Figure 2f,g) are amongst the influential factors for the flood events [20,23,32], and road and river were collected from https://www.data.qld.gov.au/dataset/baseline-roads-and-tracks-queensland (accessed on 30 May 2021) and calculated using Euclidean distance analysis [36].

The Water-Related Factors

The hydrological factors such as SPI (Figure 2h) and STI (Figure 2i) are considered important factors reflecting the soil moisture and influencing the flood events [37]. SPI measures the water erosion power resulting from the water flow to be calculated by Equation (2) [18,36].

$$\text{SPI} = \ln(A_s \cdot \tan\beta) \tag{2}$$

The lower value of this index points out the higher sensitivity to the flood hazard [3]. Moreover, the erosion and deposition of an area can be estimated by STI as follows [35]:

$$\text{STI} = \left(\frac{A_s}{22.13}\right)^{0.6}\left(\frac{\sin\beta}{0.0896}\right)^{1.3} \tag{3}$$

Similarly, the lower value of STI indicates the higher probability of the flooding. Another water-related factor was fainfall data (Figure 2j), which were obtained from meteorological stations available in the Bureau of Meteorology website (http://www.bom.gov.au/climate/data, accessed on 30 May 2021), and then they were interpolated to create a continuous climatic map.

Geological Factors

The soil type map in 1:250,000 scale (Figure 2k) and lithology in 1:100,000 scale (Figure 2l) were downloaded from the CSIRO and Australian government websites, and then the subset area was selected. According to the soil types represented in Table 2 (in 10 classes), the area is mainly covered by hard acidic yellow and red mottled soils. From a lithology point of view, the region was classified into nine categories (Table 1) predominantly in the Bundamba Group, Landsborough sandstone, Neranleigh-Fernvale beds, Bunya Phyllite, and paleocene–oligocene sediment classes.

**Table 2.** Soil contents and classes in the study area.

| Class | Description |
|---|---|
| Cd3 | Sands (Uc2.12) and siliceous sands (Uc1.21 and Uc1.22) on sandstones, grey cracking clays (Ug5.23) on shales, and shallow red clays (Uf6.12) on basalt |
| Fu2 | Shallow and stony leached loams (Um2.12) and also (Um5.2) loams. |
| Fu3 | Shallow and stony leached loams (Um2.1), and also (Um5.2) loams. |
| Kb28 | Moderate and shallow forms of dark cracking clays on the slopes. |
| MM9 | Brown and grey cracking clays (Ug5.34), (Ug5.39), and (Ug5.2), which occur on the third terrace with (Gn3.21), (Dy3.41), and (Dy3.13) soils. |
| Mw30 | Red earths (Gn2.14) with associated areas of red friable earths (Gn3.11). |
| Pl1 | Hard acidic red and yellow soils (Dr3.41), (Dr2.41), and (Dy3.41) with some areas of (Dy3.43) and (Dr3.43) soils. |
| Sj12 | Hard acidic yellow and yellow mottled soils (Dy2.41) and (Dy3.41) with (Dd1.41) on the flat areas, together with leached sands (Uc2.33 and Uc2.32) on low broad sandy banks. |
| Tb64 | Hard acidic yellow (Dy3.41) and red (Dr3.41) mottled soils. |
| Tb65 | Hard acidic and neutral yellow and red soils (Dy3.41), (Dy3.42), (Dr3.41), and (Dr2.12) on sandstones. |

Land Use

Additionally, a land use/cover map of the region (Figure 2m) was prepared from the Queensland land use mapping program (QLUMP) and reclassified into eight classes. Residential, conservation and natural environment, and intensive uses (mining) were the most dominant land cover classes in the area. Eventually, all of the datasets transformed into the same resolution of $5 \times 5$ m, and the classifications were performed using a natural break in ArcGIS 10.2 [3].

## 3. Methodology

### 3.1. Overview

Figure 3 is the workflow for this research. After data preparation (flood inventory and conditioning factors), the training and test data values of the 13 conditioning factors were extracted from the flood location points in ArcGIS. Subsequently, the coefficients of the flood conditioning factors were calculated for the variance inflation factor (VIF) analysis and tolerance. Next, the two ML models (ANN and DLNN) and an optimized DLNN via PSO were trained to categorize susceptible zones into five probability classes: very low (less than 0.2), low (between 0.2–0.4), moderate (between 0.4–0.6), high (between 0.6–0.8), and very high (more than 0.8). Finally, validations were carried out based on (1) sensitivity and specificity, (2) receiver operating characteristics (ROC) curve, and (3) TSS tests.

### 3.2. Multicollinearity Analysis

To predict the flood-prone zones, a large dataset including several causative factors needs to be modeled. Therefore, the presence of redundant data might lead to heavy and timely calculation and even lower performance [38]. Commonly, the intra-relationships between the factors and multicollinearity can be identified by VIF and coefficients of tolerance (Tolerance) [39,40]. VIF is calculated according to Equation (4):

$$\mathrm{VIF} = \frac{1}{1 - R_i^2} \qquad (4)$$

where $R_i$ is the multi correlation coefficient of *i th* factor on the remaining factors [41]. The coefficients of tolerance are also estimated from the Equation (5):

$$Tolerance = 1 - R_i^2 \qquad (5)$$

The multicollinearity among the variables is defined by a VIF value greater than 5 and the tolerance value less than 0.1 [41] and highlight the presence of linear correlation within the factors, and removal should be considered [42].

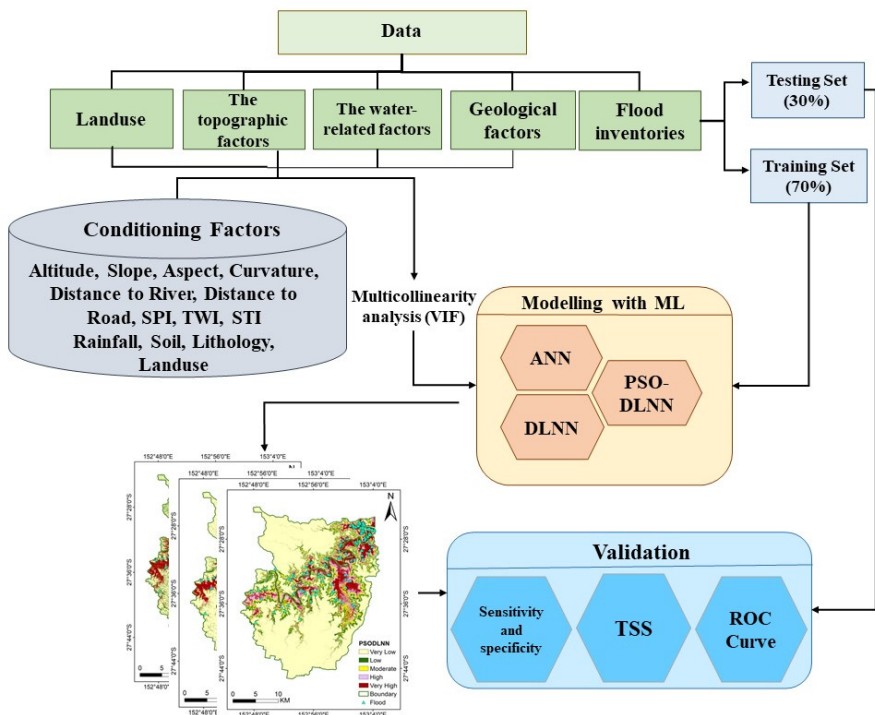

**Figure 3.** Procedures for flood susceptibility mapping in this study.

### 3.3. Modeling with ML Methods

ML algorithms have shown their potential to handle complex and non-linear problems existing in natural hazard prediction such as flood susceptibility [15,24]. With the ability of large dataset modeling [43], ML methods learn from initial so-called training datasets to entirely model, classify, and predict the variables in the whole datasets [1]. Moreover, the integration and optimization of ML techniques reflect promising results as the optimized values and parameters will enhance the algorithm performance [22]. For the present research, ANN, DLNN, and optimized DLNN via PSO were exploited using R programming with CARET packages.

#### 3.3.1. Artificial Neural Networks (ANN)

ANN as a mathematical model with the simulation capability and pattern recognition similar to the human brain can be trained by the variables [44]. It deploys a nonlinear function that literately learns the complex relationship between variables and training datasets in a network structure [45,46]. The common ANN (Figure 4) includes an input layer (with several neurons), hidden (internal) layer, and output layer where the hidden and output layer neurons multiply each input with specific weight function and weights errors are continuously calculated between the variables and corresponding observations [47]. The input layer (feed-forward of inputs) with 13 nodes of flood conditioning factors (Figure 4) was fed into ANN, and the weights of each factor in the hidden layer were calculated by the back-propagation training algorithm to iteratively minimize and adjust the error between the predictions and training datasets. The output (*Out*) form input and hidden layer can be determined as follows [25]:

$$Out = f(\sum_{j=1}^{n} w_j x_i + \theta_j) \tag{6}$$

where $f$ is a transfer function, $w_j$ defines the weight vector, and $x_i$ is the node flow (causal factors) from the inputs. $\theta_j$ represents a threshold value or bias. The transfer function

$f(x)$ is typically defined by $\left(1 + e^{-2x}\right)^{-1}$, and $w_j$ is determined by minimizing an error measure of the fit between the output and the target feature [48].

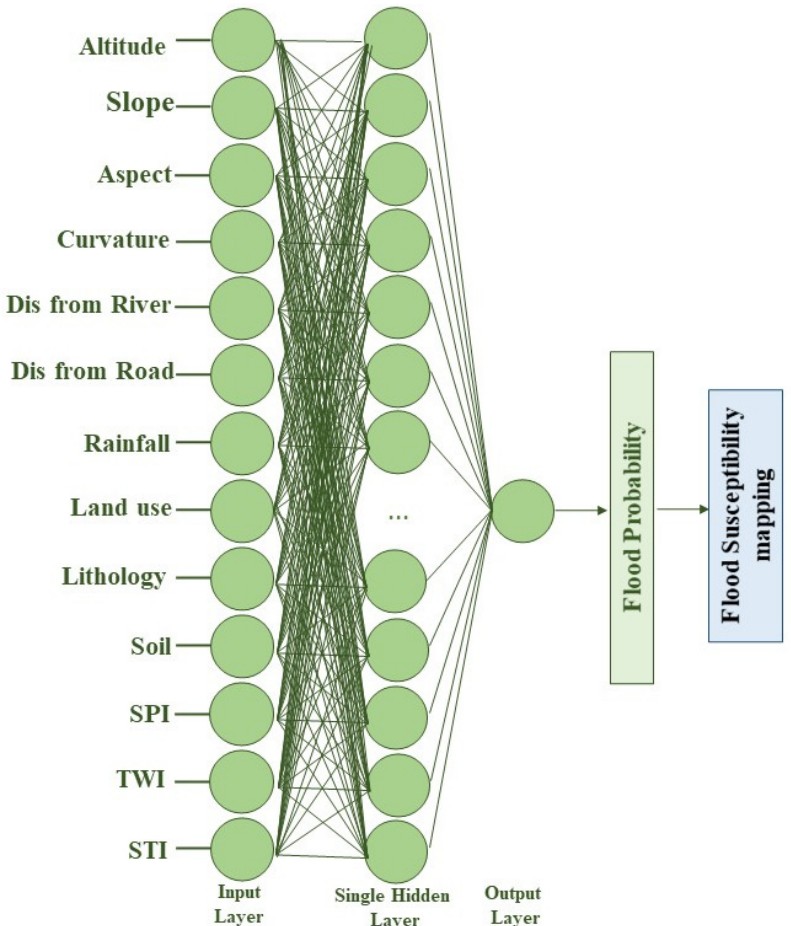

**Figure 4.** Simplified representation of ANN.

The learning rate was adjusted to 0.01 with the initial weights varying between 0.1 and 0.3, randomly. As the stopping criterion, the number of epochs was also set to 1000 with the RMSE of 0.05. During training of ANN, the final weights between layers were calculated as the level of contribution (significance) of each of the 13 factors to predict the potential of flood events. Eventually, the weights were applied to the region, the output layer was the susceptibility map, and flood-prone zones were classified (equal interval) into five probability classes (very low, low, moderate, high, and very high).

### 3.3.2. Deep Learning Neural Networks (DLNN)

DLNN (Figure 5) are generally categorized as ANN algorithms but with multiple (deep) hidden layers (based on the complexity of the features) applying feed-forward network for the back-propagation training algorithm [22,49]. The use of numerous hidden layers empowers the algorithm to better describe the nonlinear and complex features such as flooding [29]. Herein, the hidden layer was set to three according to the previous studies and to obtain stronger feature learning [22,29]. DLNN is a type of neural network with the Sigmoid function deployed within each neuron in the hidden layers to perform the back-propagation and weighting system. The sigmoid [22] activation function $f(x)$ is defined by $\left(1 + e^{-x}\right)^{-1}$. Due to training via the gradient-based algorithm with backpropagation, ReLU is used to avoid dispersing gradient [29].

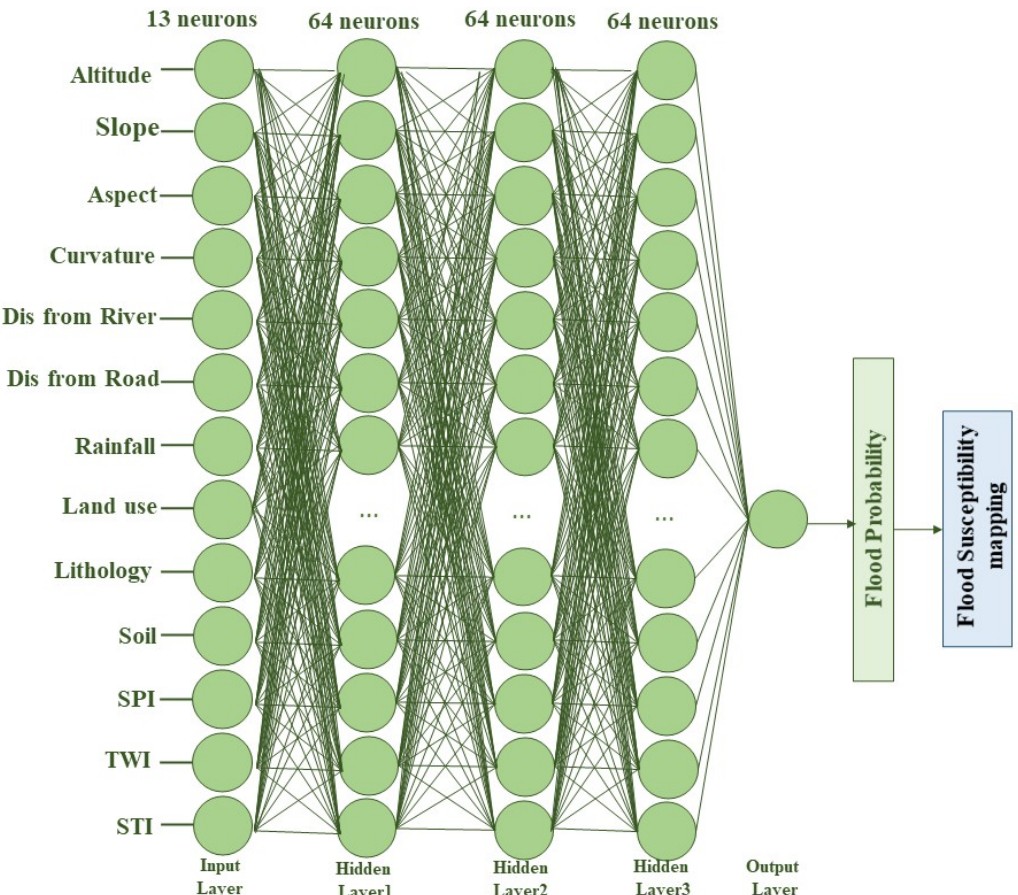

**Figure 5.** Simplified representation of DLNN.

### 3.3.3. Optimized DLNN via PSO

PSO is a computational method based on simulation of social behavior and arrangement of the insect/bird group involving a random initialization of population (swarm) of particles (encoded candidate solutions) in the search space [22,50,51]. The PSO model ironically examines the best routes and optimized solutions for the population to find the food [39]. Several features with various textural, spectral, and geometrical characteristics are selected by the best routes and optimization solutions. Random scattering of each examined feature (swarm of an individual class) formed the start of process in the searching space.

Therefore, we integrated PSO and DLNN for optimized feature selection of the datasets to obtain better performance. In the n-dimensional search space, each particle in its location moves toward the best fit solution according to its own and adjacent particles' positions [51]. The positions and velocities of each particle are dynamically saved and updated as $x_i = (x_{i1}, x_{i2}, \ldots, x_{in})$ and $v_i = (v_{i1}, v_{i2}, \ldots, v_{in})$ while the velocity is defined between $v_{max}$ and $v_{min}$ [30,51]. Then, the best position of each particle and the best position of whole group of population are recorded as $p_{best}$ and $g_{best}$, respectively [31]. The position and velocity of the particle is calculated by the following Equations [30]:

$$v_{ij}(t+1) = W \times v_{ij}(t) + r_1 \times c_1\big(p_{ij}(t) - x_{ij}(t)\big) + r_2 \times c_2\big(g_i(t) - x_{ij}(t)\big) \qquad (7)$$

$$x_{ij}(t+1) = x_{ij}(t) + v_{ij}(t+1) \qquad (8)$$

where $p_{ij}(t)$ and $g_i(t)$ are the best overall particle position and the best position of the swarm, respectively. $W$ denotes the weight of the inertia coefficient defined by $W = \frac{1}{2\ln 2}$, $c_1$ and $c_2$ are particle and swarm coefficient learning, and $r_1$ and $r_2$ stand for uniform random numbers varying between 0 and 1. The related parameters for PSO-DLNN are presented in Table 3.

**Table 3.** Parameters used in ML models.

| No | Parameter | Model | | |
|---|---|---|---|---|
| | | **ANN** | **DLNN** | **PSO-DLNN** |
| 1 | Input nodes | 13 | 13 | 13 |
| 2 | Output nodes | 2 | 2 | 2 |
| 3 | Activation | - | 'relu' | 'relu' |
| 4 | Function | - | 'Sigmoid' | 'Sigmoid' |
| 5 | reluLeak | - | 0.01 | 0.01 |
| 6 | Eta | - | 0.8 | 0.8 |
| 7 | Hidden layer unit | 1 | 3 | 3 |
| 8 | Iteration | | 500 | 500 |
| 10 | Phi | - | - | 4.1 |
| 11 | phi1 | - | - | 2.05 |
| 12 | Phi2 | - | - | 2.05 |
| 13 | W | - | - | 0.73 |
| 14 | C1 | - | - | 1.49 |
| 15 | C2 | - | - | 1.49 |

*3.4. Evaluation Methods*

The efficiency and performances of the three models were evaluated against the popular statistical methods such as positive predictive value or sensitivity, negative predictive value or specificity, the AUC, and TSS [19,37]. The evaluation methods can be better described as follows:

$$\text{Sensitivity} = \frac{TP}{TP + FN} \tag{9}$$

$$\text{Specificity} = \frac{TN}{TN + FP} \tag{10}$$

where $TP$ as true positive and $TN$ as true negative are the number of flood points and non-flood points that are correctly identified, while $FN$ (false negative) and $FP$ (false positive) represent the number of flood and non-flood points that are not correctly identified, respectively [18]. The ROC curve is depicted by plotting Sensitivity (on the y axis) against $1 - $ Specificity (on the x axis), and the AUC is a quantitative evaluation of the predictions. The values range from 0 to 1, indicating random to perfect prediction [18].

TSS is another metric to evaluate the models from a perfect prediction to a random guessing prediction by TSS value from +1 to -1 [52]. It is calculated as below:

$$\text{TSS} = \text{Sensitivity} + \text{Specificity} - 1 \tag{11}$$

**4. Results**

Data redundancy and multicollinearity within big remote sensing data might lead to heavy calculation and accuracy loss. VIF and tolerance analysis (Table 4) revealed that there was no strong correlation among the available condition factors (no VIF and tolerance values greater than 5 and less than 0.1, respectively). Therefore, all 13 conditioning factors were fed to the three aforementioned algorithms, and susceptible areas to the flood were predicted and mapped.

The susceptibility maps resulting from three models are illustrated in Figure 6 in five classes of susceptibility, and the area and percentage of each individual class were calculated in Table 5 [52]. The map from ANN represented more salt and pepper looks (Figure 6a), and predominantly very low (56.33%) and very high (24.69%) probability classes were defined across the study area. DLNN (Figure 6b) provided a smoother appearance with the very low class (67.61%) as the dominant category followed by the very high class (19.23%). On the other hand, the optimized model (Figure 6c) mainly classified the areas as the very low (61.99%) susceptible zone, while other classes appeared almost as the same percentages, ranging from 9.53% to 11.28%.

**Table 4.** Multicollinearity analysis for linearity detection within the independent variables.

| Variables | VIF | Tolerance |
|---|---|---|
| Altitude | 4.52 | 0.22 |
| Slope | 4.1 | 0.24 |
| Aspect | 1.03 | 0.97 |
| Curvature | 1.31 | 0.76 |
| Distance from river | 2.39 | 0.42 |
| Distance from road | 2.13 | 0.47 |
| Rainfall | 2.07 | 0.48 |
| Land use | 1.59 | 0.63 |
| Lithology | 1.38 | 0.72 |
| Soil | 1.99 | 0.50 |
| SPI | 1.15 | 0.87 |
| TWI | 1.69 | 0.59 |
| STI | 4.04 | 0.25 |

**Table 5.** The area and percentage of flood susceptibility classes.

| Models | Area | Susceptibility Class | | | | |
|---|---|---|---|---|---|---|
| | | Very low | Low | Moderate | High | Very high |
| ANN | Km$^2$ | 440.2872 | 144.0198 | 2.1537 | 2.1726 | 193.0005 |
| | % | 56.33 | 18.43 | 0.28 | 0.28 | 24.69 |
| DLNN | Km$^2$ | 528.4881 | 48.6306 | 24.6753 | 29.5146 | 150.3252 |
| | % | 67.61 | 6.22 | 3.16 | 3.78 | 19.23 |
| PSO-DLNN | Km$^2$ | 484.5816 | 74.4777 | 61.4268 | 73.0179 | 88.1298 |
| | % | 61.99 | 9.53 | 7.86 | 9.34 | 11.28 |

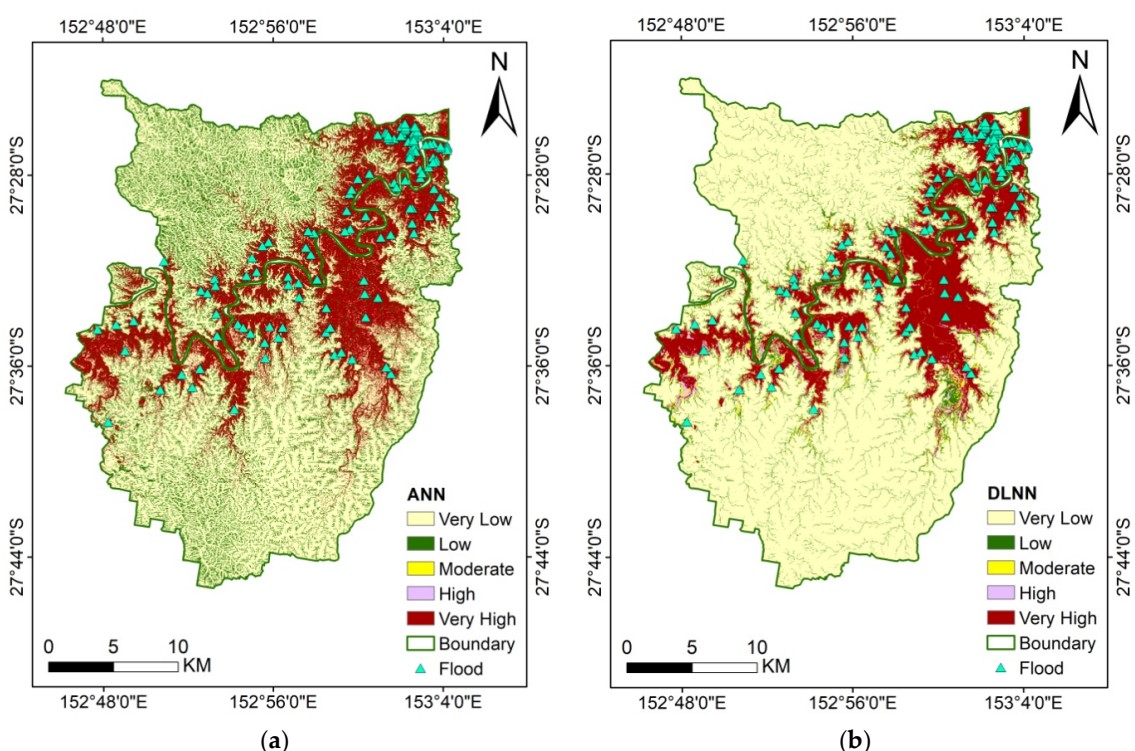

(**a**)  (**b**)

**Figure 6.** *Cont.*

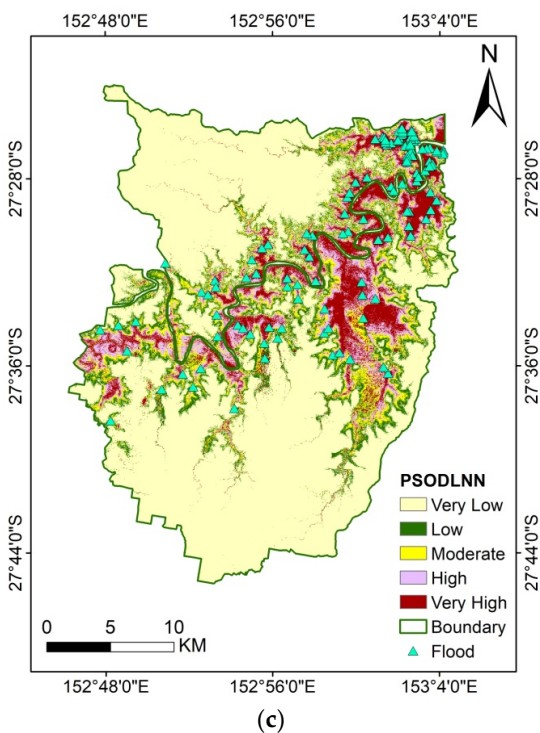

(**c**)

**Figure 6.** Flood susceptibility maps simulated using different models: (**a**) ANN, (**b**) DLNN, and (**c**) PSO-DLNN.

The percentages of flood inventory samples based on individual probability class are presented in the Figure 7. Predominantly, more than 80% of the samples were classified as "very high" probability by the three algorithms. A meaningful difference is seen in the "high" probability class by the PSO-DLNN algorithm, and almost 10% of the samples were categorized in that class. The proportions of other flood samples were minimal in other classes for every algorithm.

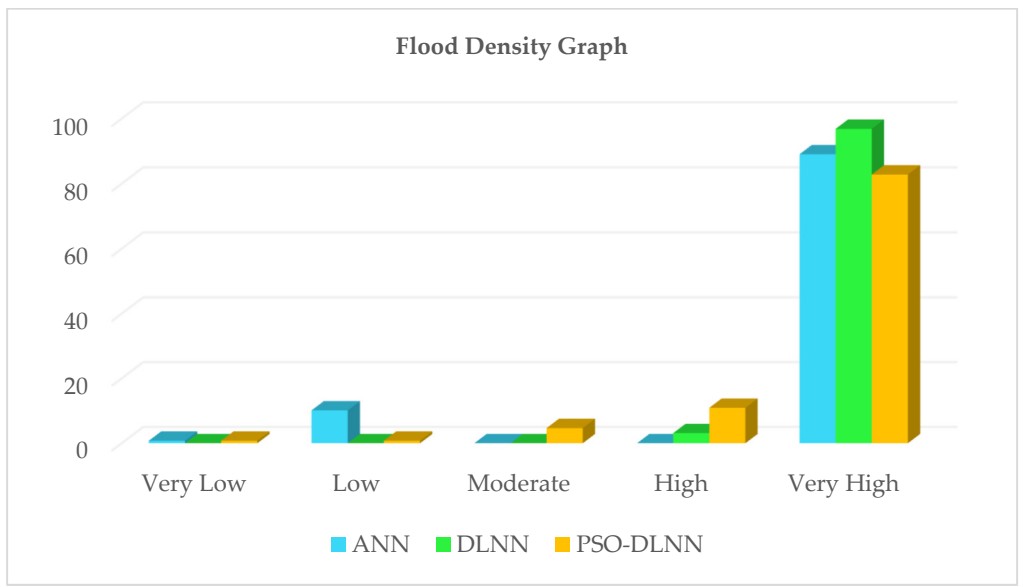

**Figure 7.** Flood density graph of ANN, DLNN, and PSO-DLNN.

Table 6 represents the evaluation of ANN, DLNN, and optimized PSO-DLNN modeling against sensitivity, specificity, TSS, and AUC metrics. The sensitivity test for the ANN model obtained the values of 0.98 and 0.94 for training and validation, respectively. Both DLNN and PSO-DLNN models slightly acquired the highest values of sensitivity (0.99)

within the training stage, while the values of 0.86 and 0.92 in the validation stage were quite lower than the one by ANN. In term of specificity, ANN showed the maximum training value (0.96) followed by PSO-DLNN (0.89) and DLNN (0.87). In contrast, in the validation stage (specificity), PSO-DLNN scored the value of 0.98, remarkably higher than those by ANN and DLNN (0.85). Again, the results of TSS for the ANN model had the maximum values (0.94) at the training level, while PSO-DLNN performed as the best in the validation stage (0.90). In addition, PSO-DLNN performances in both training and testing datasets achieved the best accuracies by the AUC metric followed by DLNN and ANN, and there was no sign of overfitting in the models. Figure 8 illustrates the AUC plot for each model. The relationship between sensitivity or true positive rate (TPR) and 1-specificity or false positive rate (FPR) can be better interpreted by these graphs.

**Table 6.** Predictive capability of ANN, DLNN, and PSO-DLNN models using train and test dataset.

| Models | Stage | Evaluation Tests | | | |
|---|---|---|---|---|---|
| | | Sensitivity | Specificity | TSS | AUC |
| ANN | Train | 0.98 | 0.96 | 0.94 | 0.98 |
| | Validation | 0.94 | 0.85 | 0.79 | 0.93 |
| DLNN | Train | 0.99 | 0.87 | 0.86 | 0.98 |
| | Validation | 0.86 | 0.85 | 0.71 | 0.96 |
| PSO-DLNN | Train | 0.99 | 0.89 | 0.88 | 0.99 |
| | Validation | 0.92 | 0.98 | 0.90 | 0.98 |

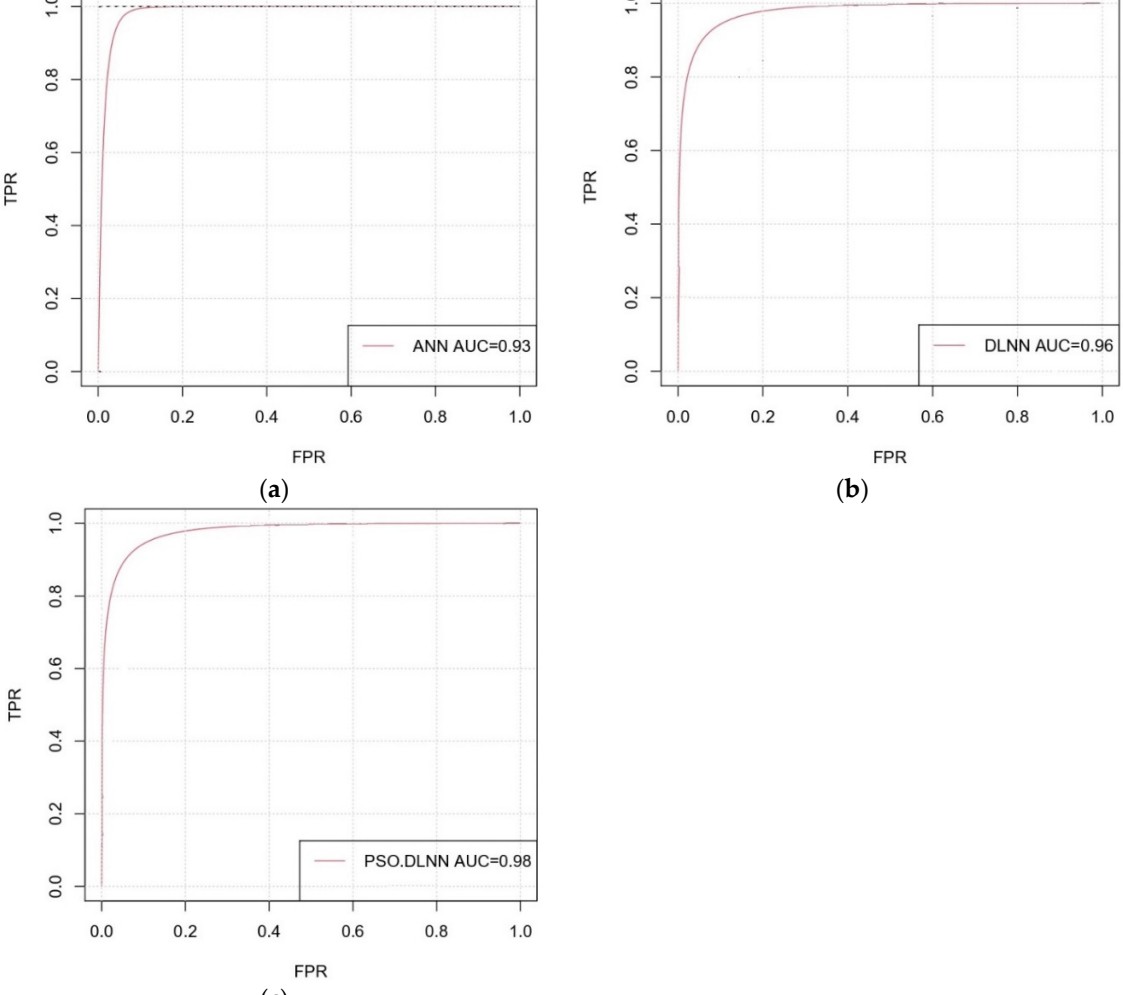

**Figure 8.** Evaluation of the flood susceptibility maps based on the AUC test (**a**) ANN, (**b**) DLNN, and (**c**) PSO-DLNNO.

The results of variable importance for PSO-DLNNO are provided in Table 7. Accordingly, the most important factor was altitude, followed by the distance from the river. STI, slope, and distance from the road were the third, fourth, and fifth ranked in terms of their significant effects on the flood occurrence in the region. Additionally, SPI seems to have no contribution to the susceptibility mapping and was ranked 13th.

**Table 7.** Variable importance analysis derived from PSO-DLNN model.

| Variables | Importance |
|---|---|
| Altitude | 100 |
| Slope | 33.05 |
| Aspect | 1.32 |
| Curvature | 16.55 |
| Distance from river | 55.44 |
| Distance from road | 29.21 |
| Rainfall | 9.31 |
| Land use | 22.63 |
| Lithology | 11.29 |
| Soil | 1.74 |
| SPI | 0 |
| TWI | 18.77 |
| STI | 39.69 |

## 5. Discussion

The availability of 10-m spatial resolution DEM in the study area along with other influential factors led to the creation of a massive geodatabase with large datasets. The robustness and high performance of the predictions were demonstrated by ANN, DLNN, and PSO-DLNN models against all accuracy assessment metrics as all the values were placed at a high level of accuracy, certainty, and goodness-of-fit. Accordingly, ANN slightly demonstrated its higher ability at the training stage to compare with other models; however, it underestimated the optimized modeling during the validation. The best AUC result was obtained by the PSO-DLNN model, followed by DLNN and ANN. TSS on validation data represented perfect agreement (rather than random guessing) in the correct prediction by PSO-DLNN model almost 10% to 20% higher than ANN and DLNN, respectively. Comparatively, it evaluated the optimized model as a reliable method for flood susceptibility mapping. Our finding is in agreement with Sachdeva et al. [19], who claimed the optimization of SVM via PSO outperformed other methods such as NN, LR, and RF. In a profound review paper, [13] highlighted that the majority of ML algorithms use a hybrid or ensemble model to improve their prediction accuracy. We also found that the optimization could enhance the ability of the ML algorithm; it reflects the potential and flexibility of ML algorithms in boosting using various methods.

Visually inspection of the susceptibility maps, the category of very high susceptible zones tended to be modeled very similarly by ANN and DLNN; however, the estimation in PSO-DLNN showed a different appearance. It highlighted the different structure of the integration model where PSO optimized the procedures and led to higher accuracy and certainty. Additionally, the use of ML models was practically effective to predict the susceptible areas to the hazard, and it was in agreement with Band et al. [39] in modeling and predicting. Figure 9 shows areas with very high sensitivity to floods. In this figure, most of the areas that considered highly sensitive to flood hazards were along rivers. All three models used in this paper predicted similar points as flood-sensitive areas. The results showed that the ANN model classified more areas as very highly sensitive to flood hazards, and the DLNN model optimized with PSO due to higher accuracy showed fewer areas as flood-sensitive areas with higher accuracyflood-sensitive. DLNN and PSO-DLNN are suitable when a larger number of samples or big data are available. These algorithms are able to estimate the results with optimum accuracy. On the other hand, the traditional ML algorithm such as ANN is not capable of handling a large number

of samples, and the outcome from this perspective is less optimal compared to the deep learning framework [39].

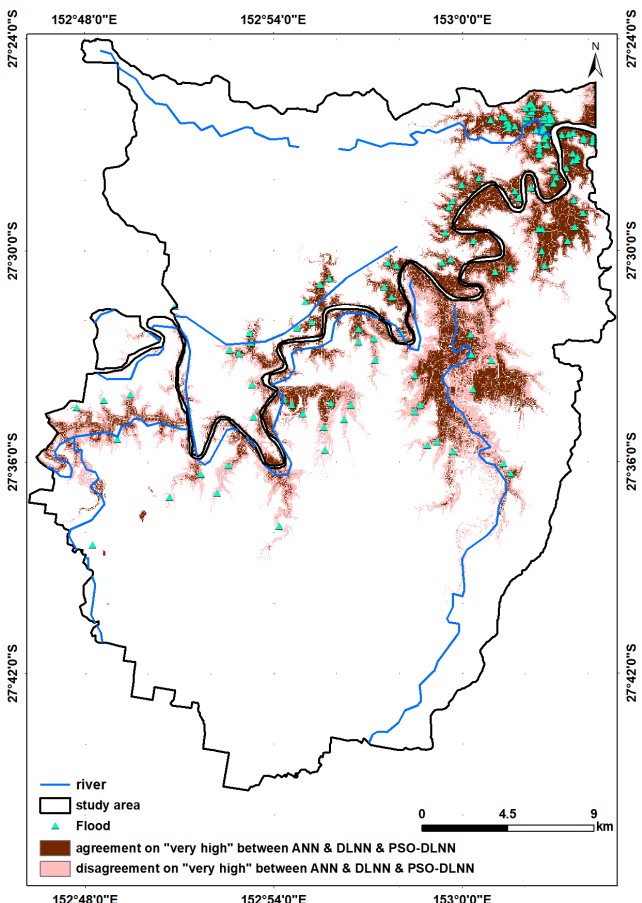

**Figure 9.** Agreement and disagreement flood susceptibility for the "very high" class simulated by ANN, DLNN, PSO-DLNN.

Figure 10 compares the flooded areas predicted by PSO_DLNN methods and determined by the hazard map. The west part of the map (Figure 10) mainly represented the same regions, while in the eastern part, the flooded areas were more expanded by the PSO_DLNN method (to compare with the hazard map). Comparisons of number of pixels and the areas labeled by the susceptibility map (PSO_DLN) and hazard map were represented in Figure 11a,b. The distribution of pixels (72,500 out of 82,768) and areas (2,175,000 out of 2,483,040 m$^2$) were mainly within "high" and "very high" classes, showing a promising agreement between the susceptibility map and hazard map. The flooded region and simulated spatial assessment of flood susceptibility using PSO-DLNN are presented in Figure 12.

The importance of altitude, distance from river, STI, and slope were consistent with the former studies [3,15,24,32]. SPI measurement showed almost equal erosion power from water flow in the entire region, and it might be the reason for the zero value in variable importance analysis by PSO-DLNN model; consequently, the regions with the high SPI values had lower sensitivity to flooding, and the probability was very low. The insignificance of SPI and aspect were also in agreement with [3,24]. Although the calculation of very high SPI values for this specific region led to rank SPI as the least important factor during the PSO-DLNN model, it cannot be interpreted as an unimportant factor, denying the effect of SPI to identify flood-prone areas in other regions. Surprisingly, the corresponding weight for rainfall revealed that this factor is not as important as other factors in this subtropical region, and our finding was consistent with [24] in another subtropical region. In this

study, TWI was not amongst the top influential factors, which were opposed to a study by [53]. The different ranking for the influential factors by several studies proved the site dependency of significant factors to intensify the flood events in each study area.

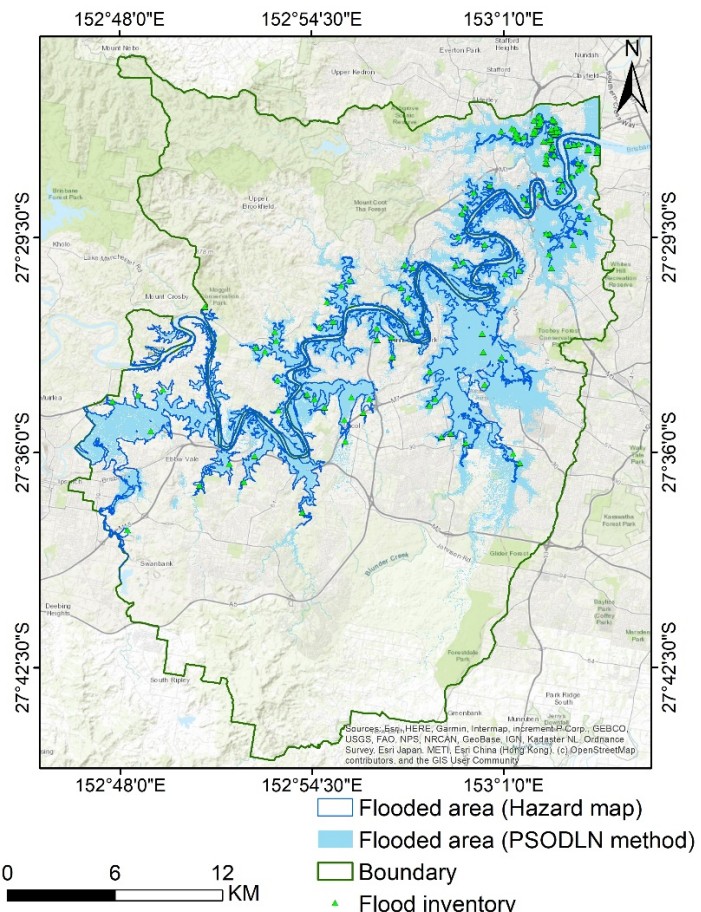

**Figure 10.** Comparison of flooded area predicted by PSO_DLNN method and hazard map.

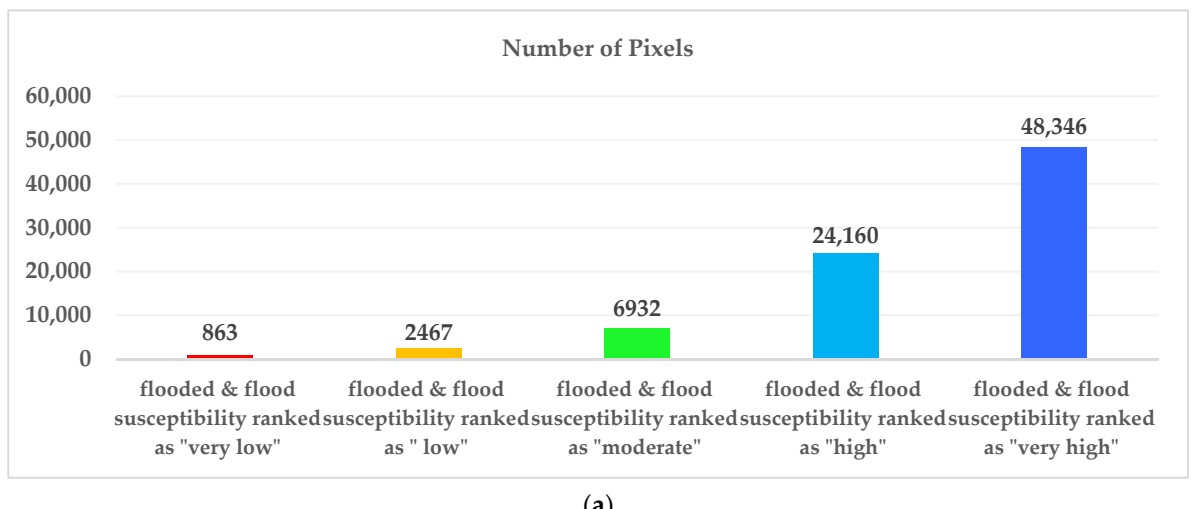

(**a**)

**Figure 11.** *Cont.*

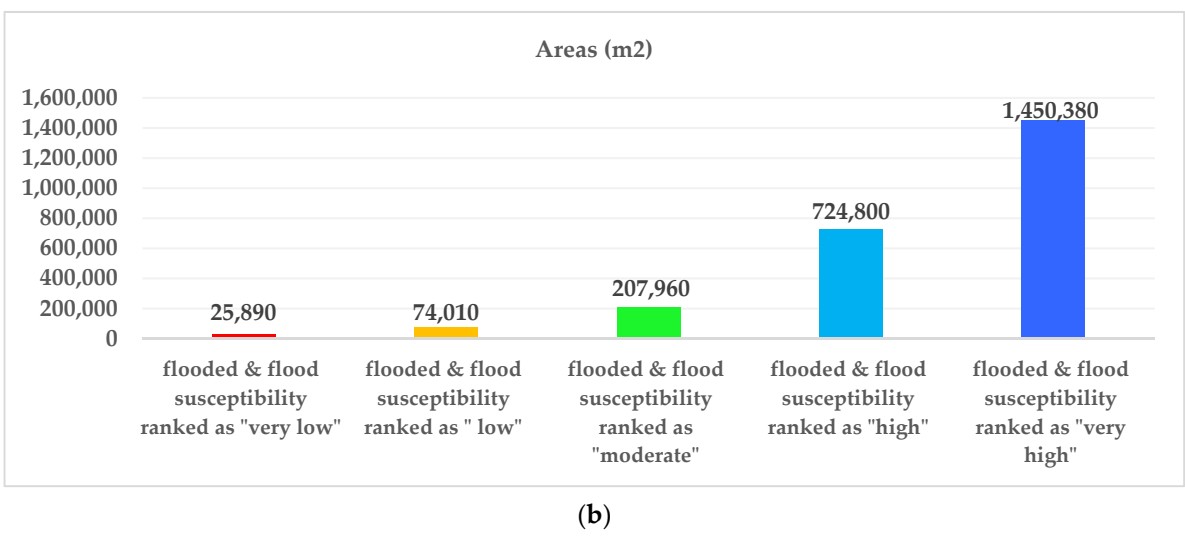

(**b**)

**Figure 11.** Comparison of flooded by hazard map and by flood susceptibility (PSO-DLNN) ranked by each class based on (**a**) number of pixels and (**b**) area (m$^2$).

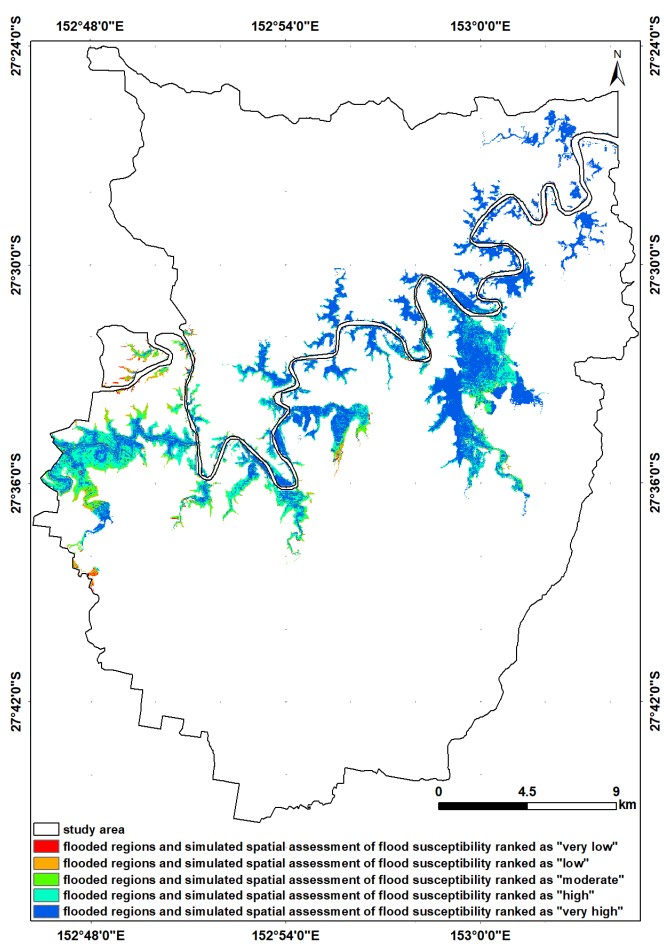

**Figure 12.** Comparison of flooded region by hazard map and simulated spatial assessment of flood susceptibility ranked by each class using PSO-DLNN.

Mostly, the areas with altitude lower than 20 m were more susceptible to flood occurrence where geologically Middle to Late Triassic volcanic units, Quaternary alluvium and lacustrine deposits, Neranleigh-Fernvale beds, and Bunya phyllite were presented, while the Bundamba group and Landsborough sandstone were less prone to flooding. From the

land-use perspective, the natural environment and conservation areas with dense forest were the least susceptible regions to flooding. On the other hand, the most susceptible areas are among the dense residential areas and intensive uses. The most probable soil types for flood events were MM9 (brown and grey cracking clays), Tb64 (hard acidic yellow and red mottled soils), and Sj12 (hard acidic yellow and yellow mottled soils) classes. Availability and accessibility to the data at the time of the flood and right after that (with high spatial resolution and accuracy) might be considered the limitation for such a dynamic hazard for better analysis and precise prediction. The final result of our research implied a form of new knowledge such as urban informatics [10] in the technique, visualization, and representation of flood hazards for efficient interpretation in the urban area. This flow of information from big data and successful prediction of potential hazardous zones might improve land use planning and economic futures.

### 6. Conclusions

To reduce potential damage and losses of future floods, flood susceptibility maps are the practical way to identify the hazardous areas in which early warning, evacuation, mitigation, and limitation for urbanization growth can be set. The availability of remote sensing and earth observation big data creates a platform for better understanding and modeling of complex phenomena such as floods. Fast and accurate analysis, visualization, and information extraction from big data are essential in natural disaster management and urban planning, which seem viable through optimization and ML algorithms. Apparently, the significance of the influential factor analysis for the region revealed that rainfall was not as important as altitude, distance from river, STI, slope, distance from road, and land use. Although the heavy rainfall was a triggering factor for flood hazards in this subtropical climate region, it could not be blamed for the flood occurrences in this specific region. As a consequence, the topographical and hydrological factors that highly suffer from urbanization and human activities were in control of such hazards. Therefore, our finding suggests revising the policy regarding urban growth and deforestation in similar regions to decrease human loss. Additionally, the precise data modeling and accurate susceptibility map by the integration of the PSO-DLNN model would enlighten the researchers and governmental sectors for proper decision making and flood management in the Brisbane river catchment. The future work will be based on creating a more accurate and complete geodatabase including other conditioning factors for flood hazards and comparing different optimization methods.

**Author Contributions:** B.K. and F.S. (Farzin Shabani) acquired the data; B.K., S.J. and F.S. (Farzin Shabani) conceptualized and performed the analysis; B.K. and V.S. wrote the manuscript and discussion and analyzed the data; N.U. supervised the funding acquisition; B.K., F.S. (Farzin Shabani), K.A. and F.S. (Fariborz Shabani) provided technical sights, as well as edited, restructured, and professionally optimized the manuscript. All authors have read and agreed to the published version of the manuscript.

**Funding:** This research was supported by the RIKEN Center for Advanced Intelligence Project, Disaster Resilience Science Team.

**Data Availability Statement:** The data used to support the findings of this study are available from the corresponding author upon request.

**Acknowledgments:** The authors would like to thank the RIKEN Centre for AIP, Japan, for providing all facilities during the research.

**Conflicts of Interest:** The authors declare no conflict of interest.

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
