# Peer review of "Deep Neural Network Utilizing Remote Sensing Datasets for Flood Hazard Susceptibility Mapping in Brisbane, Australia"

_remotesensing, doi:10.3390/rs13132638_

Round 1
Reviewer 1 Report
The paper is written in a quite mechanical way. It reads like comparing 3 models on some data (any data can be used) and a specific case description. The paper should be more theoretical and applicable in discussing how to adapt these big data methods to flood studies, and what data fit what model, etc. More important, it should also discuss the broad field of using big data to study urban sustainability. Abstract line 25: The data should not be listed in the abstract. Please be concise and summarize the key types of data used.Line 71: “remote sensing (RS) technologies and geographic information system (GIS) tackles the spatial”— I agree that this is important. But what is this paper’s novelty and contribution?
Line 114: This paragraph should be segmented into multiple paragraphs with different prediction topic areas to improve readability. Figure 1: All colors and elements in the figure need legend. Please remove irrelevant background or adding legend for all colors. Line 264: “classifications were done in ArcGIS 10.2” – not a good description. Please specify the method rather than a commercial software. The study method should be able to replicate without a commercial software. Equation 7-9: I do not think these are necessary. They are textbook equation rather than model specification of your application. Line 414: “The susceptibility maps resulting from three models are illustrated in Figure 6 in five classes of susceptibility:” – the classification scheme should be introduced in the method section. Line 488: “Broadly, the north and south 488 parts of the study area were dominant by low and very low susceptibly classes using all ML models” – discussing specific locations in your study site is not of interests to broader readership. Please interpret the prediction with the model’s characteristics. Which one is suitable for what situation and why? Line 564: “This study emphasized on the site-specific variables and 564 conditioning factors for the potential flood occurrences in Brisbane catchment, Australia” – as a paper that compares applicability of modeling methodology, too much focus on site specificity decreases your generalizability and potential impact.
Author Response
Dear Reviewer,
Thank you for your comments. Please find the attached file for our response.
Best regards,
Dr. Bahareh Kalantar

Reviewer 2 Report
The author improved the manuscript as comments suggested by Reviewer (s). Now, the manuscript can be accepted.
Minor corrections:
Please make sure all dash, font size, space, a hyphen, en dash, and capital words would be appropriate throughout the manuscript.
Please make sure the font size in the figures. Need to follow journal guidelines.
Author Response
Dear Reviewer,
Thank you for your positive feedback. The manuscript was checked according the format of journal.
Best regards,
Dr. Bahareh Kalantar
Reviewer 3 Report
General comments
This article is well written and referred enough previous work. The results are useful to other researchers in this field. This manuscript is suitable to publish in the journal of remote sensing and I recommend accepting it in the present form.
Author Response
Dear Reviewer,
Thank you for your positive feedback.
Best regards,
Dr. Bahareh Kalantar